# Electrostatically cooperative host-in-host of metal cluster ⊂ ionic organic cages in nanopores for enhanced catalysis

Liangxiao Tan[1,2], Jun-Hao Zhou[1], Jian-Ke Sun ⬤ [1✉] & Jiayin Yuan ⬤ [2✉]

The construction of hierarchically nanoporous composite for high-performance catalytic application is still challenging. In this work, a series of host-in-host ionic porous materials are crafted by encapsulating ionic organic cages into a hyper-crosslinked, oppositely charged porous poly(ionic liquid) (PoPIL) through an ion pair-directed assembly strategy. Specifically, the cationic cage (C-Cage) as the inner host can spatially accommodate a functional Au cluster, forming a [Au⊂C-Cage$^+$]⊂PoPIL$^-$ supramolecular composite. This dual-host molecular hierarchy enables a charge-selective substrate sorting effect to the Au clusters, which amplifies their catalytic activity by at least one order of magnitude as compared to Au confined only by C-Cage as the mono-host (Au⊂C-Cage$^+$). Moreover, we demonstrate that such dual-host porous system can advantageously immobilize electrostatically repulsive Au⊂C-Cage$^+$ and cationic ferrocene co-catalyst (Fer$^+$) together into the same micro-compartments, and synergistically speed up the enzyme-like tandem reactions by channelling the substrate to the catalytic centers via nanoconfinement.

[1] MOE Key Laboratory of Cluster Science, Beijing Key Laboratory of Photoelectronic/Electrophotonic Conversion Materials, School of Chemistry and Chemical Engineering, Beijing Institute of Technology, Beijing 102488, P. R. China. [2] Department of Materials and Environmental Chemistry, Stockholm University, 10691 Stockholm, Sweden. ✉email: jiankesun@bit.edu.cn; jiayin.yuan@mmk.su.se

The construction of host-in-host nanoporous materials opens up a new dimension in the design of functional porous materials[1]. Besides the beauty in exquisite architectures, such hierarchically nested hosts enable control over the guest binding and inclusion of the inner host, as well as outer host cavity functionalization, and study of the molecular interactions between the two shells[2–5]. In particular, the well-designed porous composite, which consists of robust, multiple three-dimensional nested hosts, may combine the merits of both hosts to access task-specific applications, such as selective capture of guest molecules and separation with enhanced performance in comparison with that of a single host[6–8].

Porous cages bearing intrinsic molecularly discrete nanopores have attracted much attention[9–18]. Their cage surface is engineerable via rich organic reactions and allows for straightforward construction of functional porous composite materials, e.g., host-in-host composites[19,20]. To date, efforts in the synthesis of porous composite materials have been achieved by two methods, either by covalent/coordination bonds to link the cage molecules via an organic linker into a porous framework[21–29], or by physical encapsulation of cages into a pre-synthesized porous framework/membrane[30,31]. Although a few cage-based host-in-host composites have been previously made, their preparation remains a challenge, as well as controlling precise noncovalent synthesis for enhancing the communicative and cooperative dialog between inner and outer hosts, aiming to realize high-performance applications, e.g., precise binding specific guests, and substrate-selective catalysis, etc[32–34].

In this work, we demonstrate that an organic cage-based interactive host-in-host composite can be created through an ion pair-directed supramolecular assembly strategy (Fig. 1). Specifically, a cationic cage (C-Cage) being electrostatically paired with polymerizable counterions undergoes a radical co-polymerization with a crosslinker, divinylbenzene (DVB), to build up a porous network[35]. In this network, a hyper-crosslinked porous poly(ionic liquid) (PoPIL)[36–38] shell as an outer host is formed to electrostatically surround the oppositely charged cage as the inner host, forming an C-Cage$^+$⊂PoPIL$^-$ host-in-host entity. As a step further, such ion-paired dual-host entity is proven to boost the catalytic performance of the Au cluster that is immobilized inside the C-Cage, and mimics an enzyme-type substrate-sorting behavior via electrostatic inter-host interplay. Empowered by this supramolecular structure and its related distinctive advantage, dual catalysts (Au⊂C-Cage$^+$ and cationic ferrocene derivative, Fer$^+$) bearing the same charge, which electrostatically repel each other and disfavor concerted reactions, can now be entrapped by PoPIL and forced to collaborate efficiently in enzymatic cascade reactions.

## Results

**Construction of host-in-host composites and their structure characterization.** As a proof-of-concept, first we chose a cationic cage as a model template, termed C-Cage, which was synthesized via HCl-acidification of its neutral precursor, an amine cage RCC3 (see Supplementary Fig. 1)[39,40]. The C-Cage is chemically stable, and features an amenable synthesis and accessible Cl$^-$ as the counteranion. The replacement of Cl$^-$ of the C-Cage by a polymerizable 4-styrenesulfonate (SS) anion immediately formed a supramolecular assembly (termed C-Cage-SS) that precipitated out of the aqueous C-Cage solution. The quantitative anion exchange was confirmed by proton and carbon nuclear magnetic resonance ($^1$H-NMR and $^{13}$C-NMR) spectroscopy (Supplementary Figs. 10–13), which gives a stoichiometric ratio of SS to C-Cage as 12:1 and proves all 12 Cl$^-$ anions in the C-Cage were fully replaced by SS (Supplementary Fig. 10). The titration test of the C-Cage-SS with an aqueous AgNO$_3$ solution produced no precipitate, again verifying the absence of Cl$^-$. Given the versatility in chemical structure of vinyl compounds, a variety of polymerizable anions besides SS can be in principle used, e.g., a 4-vinylbenzoic acid (VB) to produce C-Cage-VB (Supplementary Fig. 2). Extending the universality of this concept, another set of ionic supramolecular assemblies bearing an anionic cage (termed A-Cage) and a polymerizable cation, e.g., 1-cyanomethyl-3-vinylimidazolium (imidaz), was designed and termed A-Cage-imidaz (Supplementary Fig. 3). As a representative material, the C-Cage-SS and its derived host-in-host polymeric composites were illustrated here, whereas the detailed characterizations of other samples were placed in the Supplementary Figures (Supplementary Figs. 20–35).

The supramolecular assemblies of the C-Cage with the reactive vinyl anion were produced via anchoring the C-Cage into the rigid hyper-crosslinked polymer network by radical co-polymerization with an equivalent mass of DVB as crosslinker (Fig. 1). The resulting host-in-host polymeric composites were termed C-Cage$^+$⊂PoPIL$^-$ (as well as A-Cage$^-$⊂PoPIL$^+$ for the anionically charged cage). Their chemical structures were investigated by $^{13}$C cross-polarization magic-angle spinning (CP/MAS) solid-state NMR and Fourier-transform infrared (FT-IR) spectroscopy (Fig. 2a, b and Supplementary Fig. 14). The broad characteristic chemical shifts at 25.4 and 37.3 ppm as well as 124.3 and 141.3 ppm are assigned to

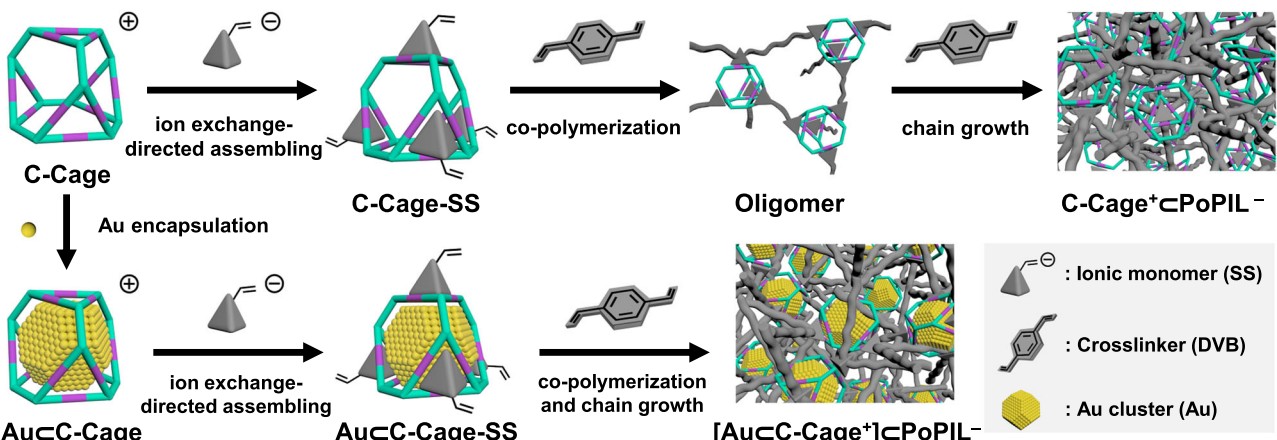

**Fig. 1 Schematic of the synthetic methodology for the hierarchical host-in-host cationic cage-porous poly(ionic liquid) (C-Cage$^+$⊂PoPIL$^-$) composites.** Note: C-Cage and 4-styrenesulfonate (SS) counteranion are selected as the representatives. Each C-Cage has 12+ charge and 12 counteranion, here, only one cation is displayed and the counteranions are not shown for the sake of clarity.

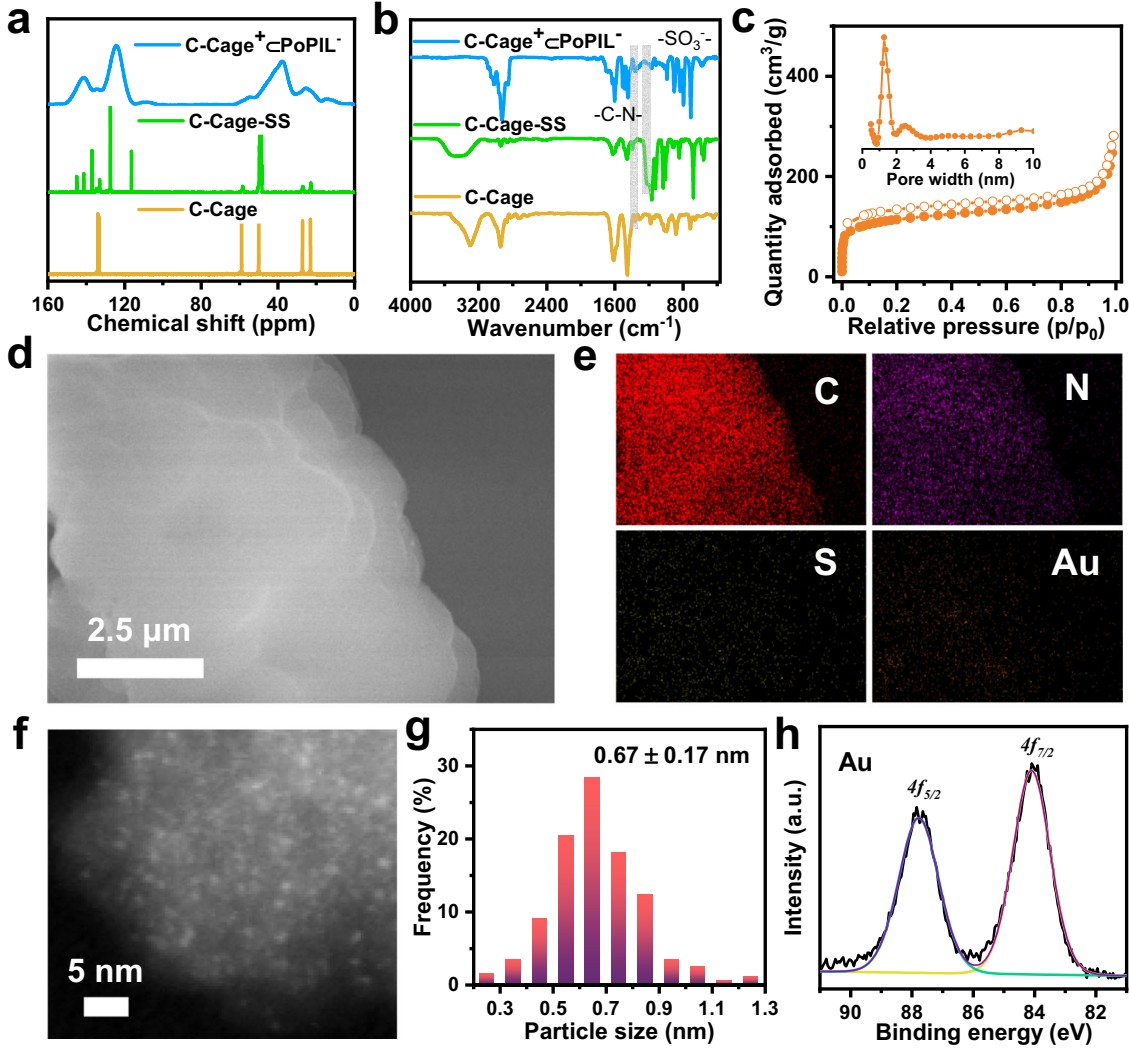

**Fig. 2 Structure characterization of the supramolecular assemblies and host-in-host composites. a** $^{13}$C NMR spectra and (**b**) FT-IR spectra of the C-Cage (yellow), C-Cage-SS (green) and C-Cage$^+$⊂PoPIL$^-$ (blue), (**c**) N$_2$ sorption isotherms and the NLDFT pore size distribution plot (inset) of the C-Cage$^+$⊂PoPIL$^-$, (**d**) SEM image and (**e**) elemental mapping images of the [Au⊂C-Cage$^+$]⊂PoPIL$^-$, scale bar: 2.5 μm, carbon element (red), nitrogen element (purple), sulfur element (gray), gold element (orange), (**f**) HAADF-STEM image of the Au clusters uniformly dispersed in the [Au⊂C-Cage$^+$]⊂PoPIL$^-$ matrix, scale bar: 5 nm, (**g**) statistic size distribution histogram of Au clusters (data calculated from 210 counts), (**h**) XPS spectrum of [Au⊂C-Cage$^+$]⊂PoPIL$^-$ showing Au $4f_{7/2}$ and $4f_{5/2}$ peaks of metallic Au.

the polymer backbone carbons and the phenyl carbons, respectively. The signals at 54.4 and 134.2 ppm as well as 145.7 ppm can be clearly identified from the cage carbons and the carbons in the SS counteranions, respectively, indicative of their successful incorporation into the supramolecular assemblies. The FT-IR spectra further champion the presence of the C-Cage and the SS counteranion, with a C-N-C stretching vibration band at 1356 cm$^{-1}$ and a broad band at 1224 cm$^{-1}$ for the sulfonate group, respectively. These results verify that the ionic supramolecular assembly structure was maintained during the polymerization process, which was next supported by its elemental analysis results (Supplementary Tables 1, 2). The formation of the host-in-host structural hierarchy was also verified by zeta potential measurements, giving a value of −37.0 ± 0.7 mV for C-Cage$^+$⊂PoPIL$^-$ (Supplementary Fig. 15a, b). It is reasonable that the nested chamber structure with the negatively charged PoPIL shell can shield the C-Cage as an inner host. Similarly, a positive value of 25.7 ± 0.6 mV was observed for A-Cage$^-$⊂PoPIL$^+$ (Supplementary Fig. 34). The micro-morphology of these supramolecular assemblies was investigated by scanning electron microscopy (SEM), which display an amorphous essence

with a rough surface agglomerated by irregular primary particles (Fig. 2d and Supplementary Fig. 16). The powder X-ray diffraction (PXRD) pattern represents only a broad peak, in agreement with its amorphous nature (Supplementary Fig. 17). The thermogravimetric analysis (TGA) results show a superior thermal stability of the polymer network till 440 °C due to its highly crosslinked state (Supplementary Fig. 18).

Analysis of the N$_2$ sorption isotherms at 77 K of C-Cage$^+$⊂PoPIL$^-$ revealed a Brunauer-Emmett-Teller (BET) surface area of 410 m$^2$ g$^{-1}$ as well as a total pore volume of 0.43 cm$^3$ g$^{-1}$ (Fig. 2c and Supplementary Table 3). The pore size distribution plot was calculated by the nonlocal density functional theory (NLDFT) modeling, indicative of a peak of 1.2 nm at the lower detection bound, as well as a broad distribution peak in the mesopore range from 2 to 10 nm (inset in Fig. 2c). Furthermore, the pore texture of the composites can be readily adjusted by varying the mass ratio of the DVB to C-Cage-SS (here the molar ratio of SS to C-Cage in the anion exchange reaction was fixed at 12:1), which revealed an ascending tendency for the BET surface areas from 410 to 647 and 836 m$^2$ g$^{-1}$ at a DVB to C-Cage-SS mass ratio from 1:1 to 3:1 and 9:1, respectively (Supplementary Fig. 19 and Supplementary Table 3).

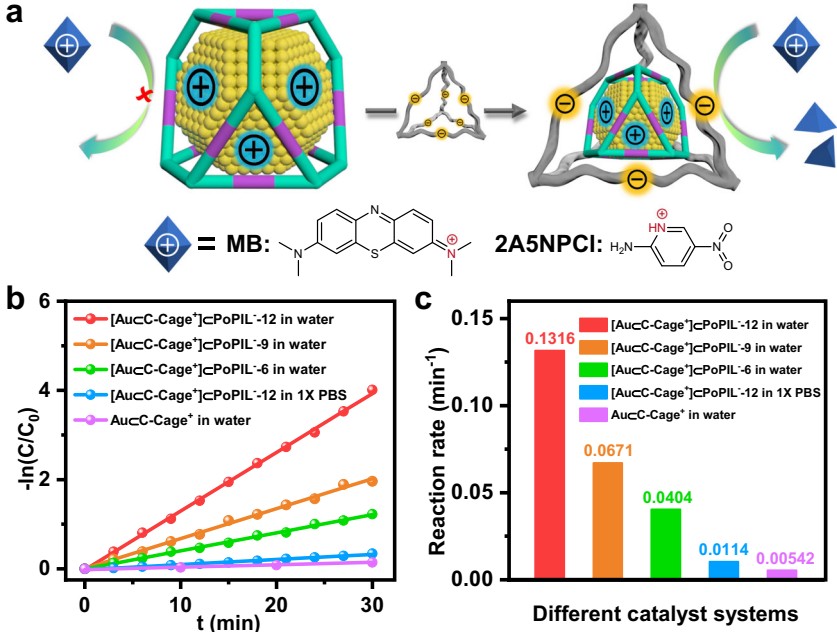

**Fig. 3 Charge-selective substrate sorting effect for a series of catalysts. a** Schematic illustration of different affinity to the cationic substrate using Au⊂C-Cage⁺ and [Au⊂C-Cage⁺]⊂PoPIL⁻ catalysts, the formulas of cationic substrates MB and 2A5NPCl are displayed (counteranions are omitted for clarity), (**b**) linear fitting and (**c**) overall comparison of the reaction rate constant for catalytic degradation of positively charged MB by Au⊂C-Cage⁺ (purple) and [Au⊂C-Cage⁺]⊂PoPIL⁻ catalysts with regulable charge density of the anionic PoPIL shell in aqueous media (The charge density can be adjusted by regulating the initial mixing molar ratio between Au⊂C-Cage⁺ and SS anion, i.e., from 1:12 to 1:9 and 1:6 in the anion exchange step prior to polymerization, the corresponding catalyst is denoted as [Au⊂C-Cage⁺]⊂PoPIL⁻-Y, Y = 12 (red), 9 (orange) or 6 (green)). The [Au⊂C-Cage⁺]⊂PoPIL⁻-12 catalyzed MB degradation in 1X PBS reaction media is also presented (blue). The above experiments were all conducted at 298 K.

Carrying the unique host-in-host materials in hand, we attempted to introduce the metal cluster (Au as a representative) into the C-Cage as inner host for catalytic applications. The synthetic procedure of [Au⊂C-Cage⁺]⊂PoPIL⁻ was similar to that of the C-Cage⁺⊂PoPIL⁻ except that the incorporation of the Au cluster into C-Cage first occurred before the anion exchange and the following polymerization steps (Fig. 1). The initial mixing molar ratio between Au⊂C-Cage⁺ and SS anion was fixed at 1:12 in the anion exchange step prior to polymerization, and the mass ratio of DVB to Au⊂C-Cage-SS was 1:1. The [Au⊂C-Cage⁺]⊂PoPIL⁻ used hereafter in the following text is based on the this stoichiometric ratio unless otherwise explicitly stated. The Au cluster in [Au⊂C-Cage⁺]⊂PoPIL⁻ was investigated by high-angle annular dark field scanning transmission electron microscopy (HAADF-STEM). Upon statistical analysis, the Au clusters possess an average size of $0.67 \pm 0.17$ nm, a value that matches well with that of the precursor Au⊂C-Cage⁺ ($0.65 \pm 0.20$ nm) (The encapsulating of Au cluster into C-Cage has been well demonstrated in our previous work[39].), indicating that the Au clusters did not aggregate during the polymerization owing to the robust nanoconfinement of the inner cage host (Fig. 2f, g). Elemental mapping visualizes the homogeneous distribution of Au element throughout the polymeric network (Fig. 2e). X-ray photoelectron spectroscopy (XPS) identified characteristic peaks of binding energies at 87.6 and 83.9 eV corresponding to $4f_{5/2}$ and $4f_{7/2}$ of metallic Au, respectively (Fig. 2h).

**Charge selective substrate-sorting effect for enhanced catalytic performance.** To explore if the ion-separated host-in-host concept can merge the merits of both C-Cage and PoPIL, the selective catalytic behavior was evaluated by [Au⊂C-Cage⁺]⊂PoPIL⁻ first as a model catalyst (Fig. 3a). All of the following catalytic experiments were conducted at 298 K, and the exact metal contents were fixed at the same amount for different catalysts

according to the ICP results (Supplementary Table 4). Since the sulfonate ligand decorated on the outer PoPIL shell is anionic and thus electrostatically attracts preferentially positive substrates such as the cationic dye methylene blue (MB, $C_{16}H_{18}ClN_3S$, molecular structure is shown in Fig. 3a) into the PoPIL host (detected by the UV-Vis absorption experiment in Supplementary Fig. 36). Upon the addition of sodium borohydride (NaBH₄), the reaction to decompose the substrate was then catalyzed by Au clusters. Since the adsorption as well as substrate diffusion may interfere, the heterogeneous system was kept stirring for 30 mins before adding NaBH₄ to fulfill the adsorption equilibrium. When dispersed in water the heterogeneous catalyst was chemically stable without detectable cage molecules leaching out of the polymer network (Supplementary Fig. 37). After introducing the substrate, the [Au⊂C-Cage⁺]⊂PoPIL⁻ catalyst was observed to show a facilitated catalytic process with ~100% degrading efficiency in 30 mins, giving a rate constant ($k$) of 0.1316 min⁻¹ (Supplementary Fig. 38b). In comparison, the nonconfined Au⊂C-Cage⁺ catalyst bearing a positively charged surface can repulse the cationic MB in solution, and displayed a slower activity with a $k$ value of 0.00542 min⁻¹, corresponding to a ~13% degrading efficiency in 30 mins (Supplementary Fig. 38d), which is 24 times smaller than that of [Au⊂C-Cage⁺]⊂PoPIL⁻. The preliminary experiments indicate that the electrostatic coupling of two oppositely charged hosts plays a decisive role in substrate-selective catalytic decomposition. Another control experiment was conducted with a similarly sized but anionic substrate methyl orange (MO, $C_{14}H_{14}N_3SO_3Na$). In this case, the nonconfined Au⊂C-Cage⁺ can degrade the substrate to ~100% with a $k$ value of 0.0749 min⁻¹, 8 times higher than that of [Au⊂C-Cage⁺]⊂PoPIL⁻ catalyst ($k = 0.00856$ min⁻¹), with which only ~41% degrading efficiency was achieved within the same time (Supplementary Fig. 39). This outcome tells a dominant charge selective substrate-sorting effect that enhanced the activity. It is noted that directly immobilizing Au

nanoparticle (with a particle size of 2.19 ± 0.52 nm, Supplementary Fig. 40) in PoPIL$^-$ without the C-Cage as inner host also showed an enhanced activity ($k = 0.0334$ min$^{-1}$) due to charge attraction between PoPIL$^-$ and the cationic MB substrate (Supplementary Fig. 41). However, the activity of Au⊂PoPIL$^-$ is lower than that of [Au⊂C-Cage$^+$]⊂PoPIL$^-$ owing to a larger size of the Au nanoparticles than that confined within the cages. It indicates the synergy and superiority of the current host-in-host system that can simultaneously confine highly active centers and meanwhile enrich the substrate around for increased activity. Moreover, the current [Au⊂C-Cage$^+$]⊂PoPIL$^-$ catalyst is robust that can be recycled for at least five times without obvious activity loss (Supplementary Fig. 42) and metal leaching as verified by the ICP test (Supplementary Table 4). The surface area calculated from the BET equation, and the HAADF-STEM measurements indicate the maintained porosity (Supplementary Fig. 43) and size of Au cluster (Supplementary Fig. 44), respectively, after catalytic cycles.

To further explore the selectivity of the current host-in-host catalysts, a competition experiment was conducted by using a mixture of equal molar amounts of MB and MO as the cationic and anionic substrates. The degradation efficiency in the presence of the [Au⊂C-Cage$^+$]⊂PoPIL$^-$ catalyst was analyzed within the same period. As observed, MB was degraded to ~92.1% within 30 min, 84 times higher than the 1.1% degradation efficiency for MO (Supplementary Fig. 45).

The Coulombic effect on the catalysis was further studied by controlling the Debye screening length (Debye length) and surface potential around the catalyst via altering the ionic strength in solution and the charge density on the outer shell, respectively. The catalytic activity of the inner Au clusters is dependent on the scale of attractive interactions between the outer PoPIL shell and the oppositely charged substrates[41–44]. Therefore, we purposely weakened such interactions via applying a higher ionic strength (e.g., phosphate-buffered saline, PBS) to the same reactions[45]. The rate constants decreased by one order of magnitude from 0.1316 to 0.0114 min$^{-1}$ when the reaction medium was changed from water to 1X PBS, while other reaction parameters kept the same (Fig. 3b, c and Supplementary Fig. 46). The effect of ionic strength on the charge screening effect was supported by calculations of the Debye length. A drop in length from ~3.06 to ~0.76 nm was obtained as the reaction medium changed (correlation of ionic strength and Debye length with reaction kinetics is given in Supplementary Table 5). Furthermore, the electrostatic interaction between the cationic substrate and the anionic outer shell was modulated by decreasing the charge density of the anionic PoPIL shell. The experiment was conducted by regulating the initial mixing molar ratio between Au⊂C-Cage$^+$ and SS anion, i.e., from 1:12 to 1:9 and 1:6 in the anion exchange step prior to polymerization (molar ratios in the resulting ion pair assemblies were well confirmed by $^1$H-NMR and elemental analysis, Supplementary Figs. 10, 12, 13 and Supplementary Table 1). The increase in the zeta potential from −38.1 to −30.9 and −26.1 mV of the resultant host-in-host [Au⊂C-Cage$^+$]⊂PoPIL$^-$ composites verified the gradual reduction of the surface charge density (Supplementary Fig. 15c, d). As such, the rate constants for the corresponding catalysts drops from 0.1316 to 0.0671 and 0.0404 min$^{-1}$ (Fig. 3b, c and Supplementary Fig. 47).

In fact, such charge-selective substrate sorting effect can be also extended to other systems, for example, the reduction of the cationic organic micropollutant 2-amino 5-nitro pyridinium chloride (2A5NPCl, C$_5$H$_6$N$_3$O$_2$Cl, molecular structure is shown in Fig. 3a, and Supplementary Fig. 48). Similarly, a much higher catalytic activity with 16 times larger rate constant (0.081 min$^{-1}$ vs 0.00501 min$^{-1}$) than that of nonconfined Au⊂C-Cage$^+$ can be obtained (Supplementary Fig. 49).

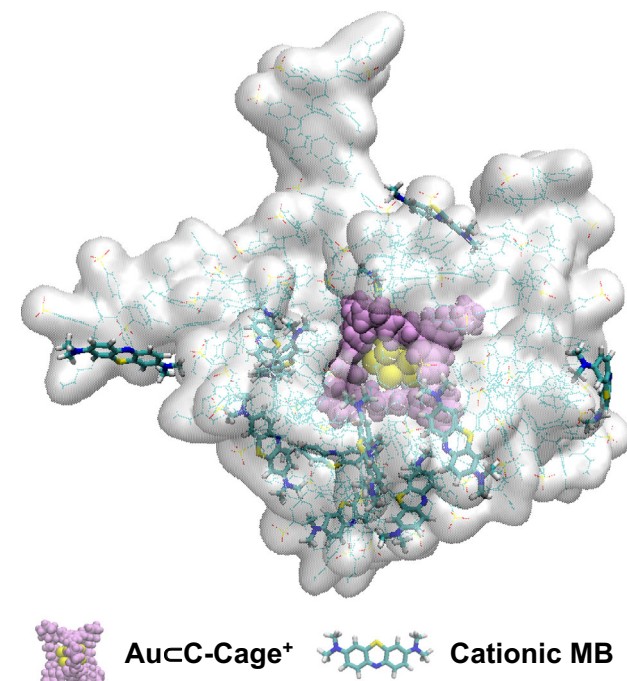

**Au⊂C-Cage$^+$**      **Cationic MB**

**Fig. 4 A snapshot of MD simulation showing the [Au⊂C-Cage$^+$]⊂PoPIL$^-$ and cationic MB substrate.** Yellow ball = Au cluster; Purple space filling mode = C-Cage$^+$; Green licorice mode = MB; White quick surface and green line mode= PoPIL$^-$. The solvent and counterions are omitted for clarity. For details of MD simulations, please see Supplementary Computational Details.

Next, the molecular dynamics (MD) simulation was conducted and reveals a higher affinity of the host-in-host [Au⊂C-Cage$^+$]⊂PoPIL$^-$ catalyst towards cationic substrates than that of nonconfined Au⊂C-Cage$^+$ catalyst owing to an attractive electrostatic force (Supplementary Figs. 62–65). It facilitates the cationic substrate to contact metal sites in confined nanosapce (Fig. 4, Supplementary Fig. 63 and Supplementary Movie 1, 2). By contrast, the anionic substrate is more difficult to reach the metal site in [Au⊂C-Cage$^+$]⊂PoPIL$^-$ catalyst than that of Au⊂C-Cage$^+$ catalyst (Supplementary Fig. 64 and Supplementary Movie 3, 4).

**Compartmentalization of dual catalysts confined in PoPIL host for enzymatic-like cascade reaction.** Since the outer PoPIL$^-$ shell is capable of selecting specific substrates into Au⊂C-Cage$^+$ containing nanochamber for accelerating the reactions, we took advantage of the [Au⊂C-Cage$^+$]⊂PoPIL$^-$ as a platform to encapsulate a dual-catalyst system in the porous polymer network. Such system consists of spatially separated Au⊂C-Cage$^+$ and cationic ferrocene co-catalyst (Fer$^+$, C$_{13}$H$_{18}$FeNCl, molecular structure is shown in Fig. 5a) within the same porous network. It mimics the compartmentalization in organelles and to catalyze a cascade reaction by channelling the substrate to nanoconfined catalysts assisted by the porous network (Fig. 5a). Here the model enzymatic-like cascade reaction is designed as follows: the glucose is first catalytically oxidized into gluconic acid and H$_2$O$_2$; next the enzymatic H$_2$O$_2$ and pre-added 3,3′,5,5′-tetramethylbenzidine (TMB) are catalyzed into H$_2$O and oxTMB[46,47]. Currently, the above-mentioned two steps were extensively explored by Au(0) and Fe(II) catalysts to mimic the properties of glucose oxidase and horseradish peroxidase,

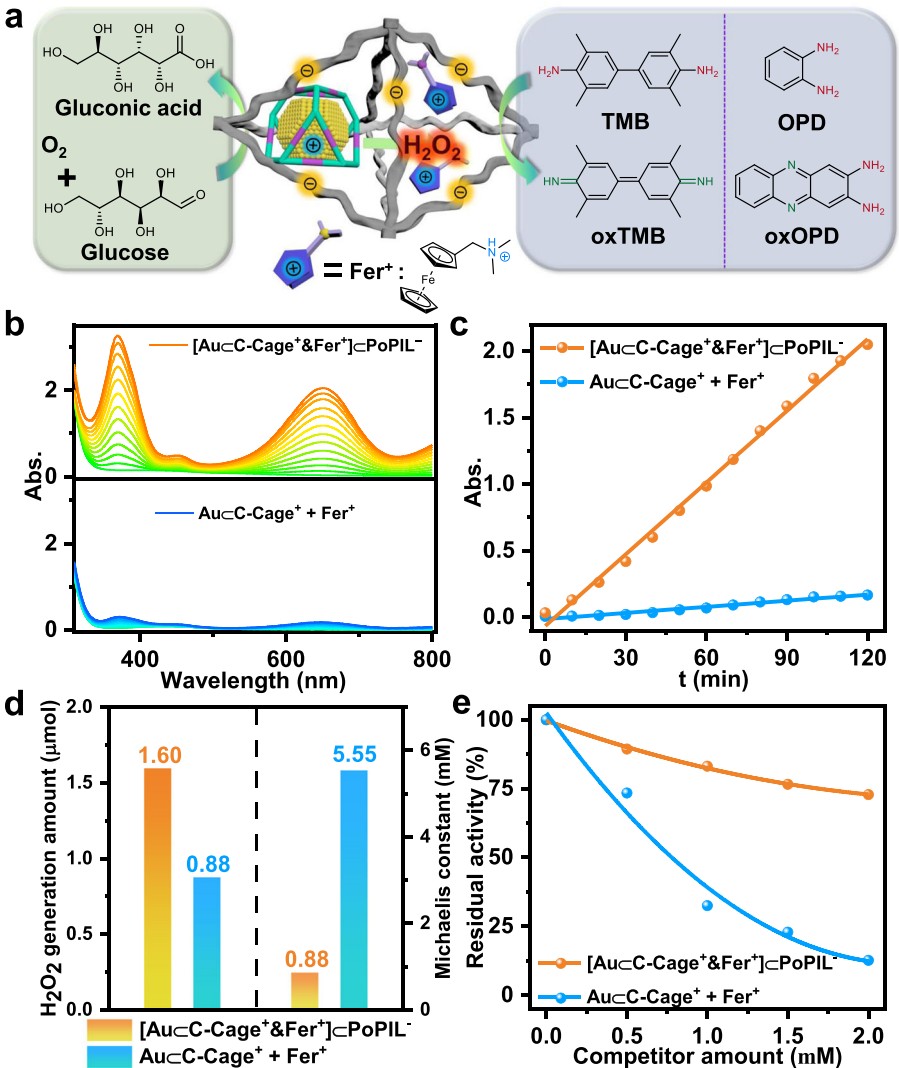

**Fig. 5 Acceleration of the enzymatic-like cascade reaction by confined [Au⊂C-Cage⁺&Fer⁺]⊂PoPIL⁻ catalyst. a** Scheme illustration of the spatially separated Au⊂C-Cage⁺ and a cationic ferrocene co-catalyst (Fer⁺) in the [Au⊂C-Cage⁺&Fer⁺]⊂PoPIL⁻ network to mimic the compartmentalization in organelles, which further catalyze enzymatic-like cascade reactions by channelling the substrate to nanoconfined catalysts assisted by the nanochamber. The molecular structure of Fer⁺ is displayed (counteranion is omitted for clarity). **b** UV-Vis spectra and (**c**) linear fitting of the reaction rate constant for TMB oxidation cascade reaction, (**d**) $H_2O_2$ production amount in the first-step of glucose oxidation and $K_m$ comparison in the second-step of TMB oxidation, (**e**) relative residual activity for the TMB oxidation cascade reaction with different competing thiourea concentrations by confined [Au⊂C-Cage⁺&Fer⁺]⊂PoPIL⁻ catalyst (orange) and a mixture of free Au⊂C-Cage⁺ and Fer⁺ catalysts (blue) in the solution. The concentrations of thiourea varied from 0 to 2 mM. The above experiments were all conducted at 298 K.

respectively[48–52]. In this case, similar to the Au⊂C-Cage⁺, the Fer⁺ catalyst was added to and spatially confined in the host-in-host porous composite. This results in the [Au⊂C-Cage⁺&Fer⁺]⊂PoPIL⁻ dual-catalyst system (detailed synthesis and characterization of the Fer⁺ as well as the [Au⊂C-Cage⁺&Fer⁺]⊂PoPIL⁻ dual-catalyst were supplied in the Supplementary Synthetic Procedures and Supplementary Figures, Supplementary Figs. 50–55), which was further evaluated in terms of its catalytic performance for the enzymatic-like cascade reaction. In a typical run, the as-synthesized catalysts and TMB-containing solution was added into the glucose solution, and the reaction was initiated by inletting oxygen. The catalytic reaction was monitored by UV-Vis spectroscopy with a maximum absorbance centered at 652 nm, indicative of a successful proceeding of the cascade reaction (Fig. 5b). The apparent steady-state kinetic assays were performed to quantify their catalytic efficiencies, in which an initial reaction rate of 0.4608 μM min⁻¹

can be obtained for [Au⊂C-Cage⁺&Fer⁺]⊂PoPIL⁻ according to UV-Vis adsorption spectra (Fig. 5c). In comparison, the homogeneous mixture of nonconfined Au⊂C-Cage⁺ and Fer⁺ catalysts delivered only a rate of 0.03923 μM min⁻¹ under the same condition, which is one order of magnitude lower than its heterogeneous counterpart [Au⊂C-Cage⁺&Fer⁺]⊂PoPIL⁻ that contains the nanoconfined catalysts.

## Discussion
To pinpoint the work mechanism of the tandem catalyst system, the activity of each step was tested to compare the confined catalyst and the free counterparts. We found that the [Au⊂C-Cage⁺&Fer⁺]⊂PoPIL⁻ can generate 1.8 times of $H_2O_2$ more than that of the mixture of free Au⊂C-Cage⁺ and Fer⁺ catalysts under the same condition in the first reaction step (1.60 μmol vs 0.88 μmol, Fig. 5d and Supplementary Fig. 56); in the second step, the enzymatic Michaelis-Menten kinetics curve with respect to

a fixed $H_2O_2$ concentration revealed that the confined [Au⊂C-Cage$^+$&Fer$^+$]⊂PoPIL$^-$ dual-catalyst can dramatically decrease $K_m$ (0.88 mM) when compared with the mixture of free Au⊂C-Cage$^+$ and Fer$^+$ counterparts ($K_m = 5.55$ mM) (Fig. 5d and Supplementary Fig. 57), indicating that the Fer$^+$ in a nano-confined space has a higher affinity to the TMB substrate[53,54]. Such enhanced activity for dual catalysts confined in nanoporous cavity is reminiscent of the substrate channelling effect in nature enzymatic cascade reaction, i.e., a process that involves the coupling of enzymatic reactions by a direct transfer of the product of one enzyme to the next one without passage through the bulk solution. This effect increases efficiencies and yields in multistep reactions involving diffusion processes between spatially separated catalytic sites[55–57]. To quantify this effect, a control experiment based on the above-mentioned cascade reaction was conducted for [Au⊂C-Cage$^+$&Fer$^+$]⊂PoPIL$^-$ and mixed catalysts (free Au⊂C-Cage$^+$ and Fer$^+$), in which the cascade intermediate was consumed in a competitive side reaction (Supplementary Fig. 58). It has been well demonstrated previously that the effect of the side reaction is descending when the effect of substrate channelling is ascending, while without the channelling effect, the cascade activity will be drastically decreased since the intermediates can be very easily depleted in the bulk environment[58,59]. In detail, an inhibitor (thiourea as a strong reductant) was added initially, which can consume the generated $H_2O_2$ intermediate as a product from the first step, thus interfering with the second step of the catalytic cascade reaction. As shown in Fig. 5e, when the cascade reaction was catalyzed by a mixture of free Au⊂C-Cage$^+$ and Fer$^+$ catalysts in the solution, a much lower residual activity of below 13% was observed at a high competing agent concentration. However, the channelling phenomenon in the assembled cascade reaction by the [Au⊂C-Cage$^+$&Fer$^+$]⊂PoPIL$^-$ catalyst may prevent inhibitors from exerting an inhibitory effect on the catalysts by restricting them from accessing the active sites of confined dual catalysts; up to approximate 73% residual activity was maintained at the same competing agent concentration (Supplementary Figs. 59, 60). Therefore, the enhanced activity in the cascade reaction in the current host-in-host system is attributed to the nanoconfinement of the PoPIL, which likely provides an accumulated concentration of the substrate inside pores around the Au clusters, and channells the substrate to dual catalysts compartmentalized by nanochamber, thus drastically enhancing the final activity in comparison to their free counterpart in the solution[60–65]. In fact, the current catalyst system is useful and can be applied to other enzymatic-like cascade reactions. For example, in another test the glucose was first catalytically oxidized and generated gluconic acid and $H_2O_2$; then the enzymatic $H_2O_2$ and pre-added o-phenylenediamine (OPD) were further catalyzed into 2,3-diamino-phenazine (oxOPD) (Fig. 5a), which showed a 10-fold activity of the simple mixture of the free dual catalysts in the solution (Supplementary Fig. 61).

In summary, a host-in-host composite has been successfully synthesized via an ion pair-directed self-assembly strategy. The inner C-Cage host can entrap Au cluster and its catalytic activity can be significantly enhanced by covering an outer PoPIL shell through a charge-selective substrate sorting effect. Such host-in-host nanochamber can even accommodate dual catalysts with repulsive charge (Au⊂C-Cage$^+$ and Fer$^+$) and compartmentalize them for better activity in the enzymatic-like cascade reactions. The present work provides an excellent platform to fabricate advanced porous materials associated with the merits of porous molecules and the tunable, highly versatile feature of functional PoPILs, which may inspire the applications in advanced catalysis, separations, drug delivery, and sensing.

## Methods

The full methods for this manuscript can be found within the Supplementary Information.

## Data availability

The main data generated in this study are provided in the article and Supplementary Information. Additional data are available from the corresponding author on request.

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

## Acknowledgements

J.K. Sun thanks for financial support from the National Natural Science Foundation of China (Grant Nos. 22071008 and 52003029), the High-level Overseas Talents Program of China, the Central University Basic Research Fund of China (3100012212106), Excellent Young Scholars Research Fund from the Beijing Institute of Technology (3100011181910). The technical support from the staff at the Analysis & Testing Center, Beijing Institute of Technology is also appreciated. J.Y. is grateful for financial support from European Research Council (ERC) Starting Grant NAPOLI-639720, Swedish Research Council Grant 2018-05351, Dozentenpreis 15126 from Verband der Chemischen Industrie e.V. (VCI) in Germany, the Wallenberg Academy Fellow program (Grant KAW 2017.0166) in Sweden, and the Stockholm University Strategic Fund SU FV-2.1.1-005.

## Author contributions

J.S. and J.Y. conceived the idea and led the project. L.T. conducted the whole experiment and data analysis. J.Z. conducted the theoretical calculations. J.S. and J.Y. wrote the manuscript with help from L.T. and J.Z. All authors discussed and revised the manuscript.

## Funding

## Competing interests

The authors declare no competing interests.
