## [Peer Review File · Nature Communications]

Electrostatically cooperative host-in-host of metal cluster U ionic organic cages in nanopores for enhanced catalysisREVIEWER COMMENTS

Reviewer #1 (Remarks to the Author):

Report on manuscript entitle 'Electrostatically Cooperative "Host in Host": Metal Cluster \subset Ionic Organic Cages in Nanoporous Poly(ionic liquid)s' by L. Tan et al. for publication in Nature Communication.

The results presented by L. Tan et al. are very innovative and open up new perspectives for the rational design of hybrid nanomaterials for applications in many fields and in particular in catalysis and separation. However, the manuscript is written in a very general way whereas the examples of encapsulated clusters and cocatalysts are limited to 1 example of cluster (Au) and to 1 example of cocatalyst (ferrocen cation). On the one hand, the experimental work is very well done leading to satisfactory structural and analytical investigations and to satisfactory catalytic activity determinations. On the other hand, from these results, the authors write a manuscript in such a way that this same manuscript can also deal with many other possible catalysts and catalyst tandem. In the end, the manuscript is not precise enough and the title is also too broad. The formula and the cluster as well as the cationic "ferrocene" must be described in the manuscript and not in the complementary part. Why are these moieties taken as an example? The same comments could be made for the cages. There is a lack of other examples of encapsulated catalysts justifying to write the manuscript as it is actually written. The manuscript and the title lack of chemical formulas. In the end, despite a very original work, the manuscript is difficult to follow because formulas are missing and too much information is in the supplementary. Therefore, I cannot recommend the manuscript for publication in Nature Communications.

Reviewer #2 (Remarks to the Author):

Yuan et al. report on hierarchical supramolecular assembly built from anionic (A-cage) or cationic (C-cage) supramolecular cages which are selectively embedded within ionic organic polymer (PoPIL for porous Poly-Ionic Liquid). The synthetic methods appears well thought and straightforward leading to a series of polymer-based compounds noted I-cage+ \subset PoPIL- through polymerisation process between ionic polymerizable monomers (SS or VB for C-cage and imidaz for the A-cage) and DVB. The authors evidence that such a hierarchical matrix is useful to design catalytic materials based on Au-particles stabilized within the cationic cage A-cage+. The catalytic results using either cationic substrate (MB) or anionic one (MO) are consistent with an enhanced activity and selectivity arising from the outer ionic host that control the delivery of the substrate at the catalytic site (Au NPs). The concept is then extended to multifunctional catalyst that contains two catalytic functions such as the Au-NPs, located within the C-cage and cationic derivative of ferrocene (Fe+). These experimental results, supported by theoretical calculations appear quite convincing and interesting enough to be published in NCOMMS. Nevertheless, some major revisions should be needed prior to any publications and the following points have to be addressed:

- 1- The author determine the kinetic constant from the kinetic experiment. They never give the corresponding temperature which should be given at each time. For instance, Figure 3 shows kinetic results where the experiments temperature is missing.
- 2- The authors carried out an experiment using PBS to vary the ionic strength. They must give the composition, especially the nature of the cation which interfere with the anionic PoPIL outer framework.

3- In the protocols of synthesis, before polymerization, a step corresponds to the formation of ionic salts between the C-cage and the SS ionic monomer. As claimed by the Authors, mixing of both species in water leads to the precipitation of the salt based on the stoichiometric ratio SS:C-cage = 12 : 1. Under such conditions, how can authors control the formation of other ratios such as 9: 1 or 6: 1 which are then used for the synthesis of C-Cage \subset PoPIL. This point should be cleared.

4- In relationship with the previous point 3), the chemical formula, the corresponding elemental analysis with found and calculated mass percentages and the molecular weight should be given for all the materials, including molecular intermediate (Cage, C-Cage, A-Cage, C-cage-SS, C-cage-DV, A-Cage-Imidaz, C-Cage \subset PoPil, A-Cage \subset PoPIL, MCCC-Cage \subset PoPil and MCCC-Cage&Fe \subset PoPil). This is absolutely required if readers were to reproduce these results. The yield should be also given. The supplementary tables 1 and 3 given in SI. 5 are not sufficient to support the presented data.

5- Actually, description of the syntheses should be more detailed in the supporting information section. For instance, in paragraph SI. 2.5, the A-cage formation results from reduction of the CC3-OH cage in the presence of NaBH₄. The procedure consists of solvent removal (methanol, DCM and water) under vacuum. Finally, the residue is washed with water. Is it enough to remove the sodium ions and the boron-containing species? What is the purity of this compound? This point should be cleared and the elemental analysis and yield should be given.

Reviewer #3 (Remarks to the Author):

I am sorry to be very ambivalent on this manuscript. On one hand, it presents beautiful science. On the other hand, the paper is not easily readable, since it is filled with many methodological details and it presents a discussion that is not always appropriate given the data presented. I appreciate the computational component of the manuscript but this is not the key one. I believe that it will be an editorial choice, it is possible that other referees see in the paper things that I missed. But I think that the paper would better fit to a more specialized journal.

Reviewer #4 (Remarks to the Author):

In this work, a dual-host supramolecular composite was created through an ion pair-directed supramolecular assembly strategy, which can be used for the immobilization of catalysts (Au cluster or cationic ferrocene). This dual-host molecular hierarchy enables a charge-selective substrate sorting effect to the catalysts, which amplifies their catalytic activity by at least one order of magnitude as compared to catalysts confined by the mono-host. This work is of fundamental importance and novelty. Considering this, it can be accepted after the following major issues are addressed:

1. How can the authors prove that metal clusters are enclosed in cages rather than adsorbed to surfaces? Related discussion and characterization should be provided.
2. How many metal clusters are present in the catalyst? How can the authors ensure consistency in the number of active centers in catalytic reactions?
3. How about the stability of the catalyst? The TEM, ICP, and BET characterizations of the catalysts after cyclic reaction should be provided.
4. Is the reaction efficiency of this tandem catalyst comparable to that of a natural enzyme cascade reaction (such as glucose-HRP system)?
5. Please check Figure 5b and Supplementary Figure 55.

6. In Figure 5e, the reduced catalytic activity may be attributed to the toxicity of the catalytic central site rather than the consumption of intermediates of H₂O₂. More detailed experimental evidence needs to be supplemented.
7. Fig. 2f is recommended to be updated because of the low magnification.

Point-by-point response to reviewers' comments

Reviewer #1 (Remarks to the Author):

1. The results presented by L. Tan et al. are very innovative and open up new perspectives for the rational design of hybrid nanomaterials for applications in many fields and in particular in catalysis and separation. However, the manuscript is written in a very general way whereas the examples of encapsulated clusters and cocatalysts are limited to 1 example of cluster (Au) and to 1 example of cocatalyst (ferrocen cation). On the one hand, the experimental work is very well done leading to satisfactory structural and analytical investigations and to satisfactory catalytic activity determinations. On the other hand, from these results, the authors write a manuscript in such a way that this same manuscript can also deal with many other possible catalysts and catalyst tandem.

Response: We thank the reviewer for the useful suggestions and comments. As suggested, in the revised manuscript, we have deleted the relevant statement of generality for the synthetic methodologies and tandem catalysis to specify the materials applications.

2. In the end, the manuscript is not precise enough and the title is also too broad.

Response: We are grateful for the reviewer of their useful suggestion. We have divided the section of "Results" into several themed parts and subtitled them with concise phrase to make the whole manuscript more readable and clearer for the readers. Moreover, the title of manuscript has been changed as follows to be more precise to reflect the content of manuscript:

"Electrostatically cooperative "host in host": metal cluster \subset ionic organic cages in nanoporous poly(ionic liquid)s for enhanced catalytic performance".

3. The formula and the cluster as well as the cationic "ferrocene" must be described in the manuscript and not in the complementary part.

Response: We thank the reviewer for the thoughtful suggestion. Since the current Au clusters are synthesized by direct reduction of the metal precursor by NaBH₄ agent, and they don't possess atom-precise structures, therefore, the formulas of the clusters are difficult to determine. However, the formula of cationic ferrocene (Fer⁺) was supplied in the revised manuscript. Meanwhile, the formulas of other compounds are also provided in the revised manuscript and supplementary information including substrates, cage molecules as well as their supramolecular assemblies for clarity.

4. Why are these moieties taken as an example? The same comments could be made for the cages.

Response: Thank you for the suggestion. The moieties we chose in the manuscript are typical models in literature. For examples, CC3 cage is one of the most widely used molecular cages in constructing functional hybrid materials, membranes and so on due to its easy-to-modify molecular architecture as well as intrinsic pores (*Adv. Mater.* **2016**, *28*, 2629; *Angew. Chem. Int. Ed.* **2019**, *58*, 2638; *Nat. Commun.* **2020**, *11*, 4927). Supported Au clusters or nanoparticles are active sites for many organic reactions, and even featuring GOx enzyme-mimicking activity (*Small* **2018**, *14*, 1803256). With high stability, ferrocene-based materials are considered to be typical Fenton catalyst to decompose H₂O₂, followed by generating ·OH radical for subsequent oxidation reaction that mimics the HRP enzyme activity (*ACS Appl. Mater. Interfaces* **2021**, *13*, 53574). As such, the

above-mentioned moieties were selected as typical components to construct hierarchical “host-in-host” catalysts.

5. There is a lack of other examples of encapsulated catalysts justifying to write the manuscript as it is actually written.

Response: We thank the reviewer as well as editor for their constructive suggestions. In the revised manuscript, we have removed the claim on the generality of the strategy to make the methodology more specific.

6. The manuscript and the title lack of chemical formulas. In the end, despite a very original work, the manuscript is difficult to follow because formulas are missing and too much information is in the supplementary.

Response: We thank the reviewer for the useful suggestion and comment. In the revised manuscript and supplementary information, we have added the chemical formulas for Fe^+ , substrates, cage molecules as well as their supramolecular assemblies for clarity.

Reviewer #2 (Remarks to the Author):

Yuan et al. report on hierarchical supramolecular assembly built from anionic (A-cage) or cationic (C-cage) supramolecular cages which are selectively embedded within ionic organic polymer (PoPIL for porous Poly-Ionic Liquid). The synthetic methods appears well thought and straightforward leading to a series of polymer-based compounds noted I-cage⁺⊂PoPIL⁻ through polymerisation process between ionic polymerizable monomers (SS or VB for C-cage and imidaz for the A-cage) and DVB. The authors evidence that such a hierarchical matrix is useful to design catalytic materials based on Au-particules stabilized within the cationic cage A-cage⁺. The catalytic results using either cationic substrate (MB) or anionic one (MO) are consistent with an enhanced activity and selectivity arising from the outer ionic host that control the delivery of the substrate at the catalytic site (Au NPs). The concept is then extended to multifunctional catalyst that contains two catalytic functions such as the Au-NPs, located within the C-cage and cationic derivative of ferrocene (Fe⁺). These experimental results, supported by theoretical calculations appear quite convincing and interesting enough to be published in NCOMMS.

Response: We thank the reviewer for the very positive comments.

Nevertheless, some major revisions should be needed prior to any publications and the following points have to be addressed:

1. The author determine the kinetic constant from the kinetic experiment. They never give the corresponding temperature which should be given at each time. For instance, Figure 3 shows kinetic results where the experiments temperature is missing.

Response: We thank the reviewer for pointing out this issue. The temperature is actually an important parameter in catalytic process. In fact, we have carried out all the catalytic experiments at 298 K. In the revised manuscript, we have added this information in the following sentences:

In the main text (1st paragraph, page8):

“All of the following catalytic experiments were conducted at 298K,...”

In the figure caption parts of Fig. 3 and 5 (page 10 and 14 respectively in the main text):

“The above experiments were all conducted at 298 K.”

In the supplementary information (3. Catalytic experiments, page 9), the following sentence has also been added:

“To eliminate the catalytic activity discrepancy caused by temperature, all the catalytic experiments were conducted at 298 K.”

2. The authors carried out an experiment using PBS to vary the ionic strength. They must give the composition, especially the nature of the cation which interfere with the anionic PoPIL outer framework.

Response: We thank the reviewer for the thoughtful suggestion and comment. The 1X PBS solution is composed of 137 mM NaCl, 2.7 mM KCl, 10 mM Na₂HPO₄ and 1.8 mM KH₂PO₄. The composition was added in the following sentence in supplementary information (3.1.2 Coulombic effect, page 9):

“...1X phosphate-buffered saline solution (PBS, composed of 137 mM NaCl, 2.7 mM KCl, 10 mM Na₂HPO₄ and 1.8 mM KH₂PO₄)...”.

3. In the protocole of synthesis, before polymerization, a step corresponds to the formation of ionic

salts between the C-cage and the SS ionic monomer. As claimed by the Authors, mixing of both species in water leads to the precipitation of the salt based on the stoichiometric ratio SS:C-cage = 12 : 1. Under such conditions, how can authors control the formation of other ratios such as 9: 1 or 6: 1 which are then used for the synthesis of C-Cage<PoPIL. This point should be cleared.

Response: We thank the reviewer for the thoughtful suggestion. The formation of different supramolecular assemblies with varied ratios (such as SS:C-Cage=9:1 or 6:1) was controlled by mixing anionic SS counterion and cationic C-Cage precursor at certain molar ratio of 9:1 or 6:1. The high purity as well as molar ratio of SS counteranion to C-Cage from 12:1 to 9:1 and 6:1 within the corresponding supramolecular assemblies were well demonstrated by ¹H-NMR measurements and elemental analysis (please refer to Supplementary Fig. 10, 12, 13 and Table 1). The following sentence has been corrected in the revised manuscript for clarity:

“The experiment was conducted by regulating the initial mixing molar ratio between Au<C-Cage⁺ and SS anion, *i.e.*, from 1:12 to 1:9 and 1:6 in the anion exchange step prior to polymerization (molar ratios in the resulting ion pair assemblies were well confirmed by ¹H-NMR and elemental analysis, Supplementary Fig. 10, 12, 13 and Table 1).” (1st paragraph, page 11 in the main text)

4. In relationship with the previous point 3), the chemical formula, the corresponding elemental analysis with found and calculated mass percentages and the molecular weight should be given for all the materials, including molecular intermediate (Cage, C-Cage, A-Cage, C-cage-SS, C-cage-DV, A-Cage-Imidaz, C-Cage<PoPil, A-Cage<PoPIL, MC<C-Cage<PoPil and MC<C-Cage&Fe<PoPil). This is absolutely required if readers were to reproduce these results. The yield should be also given. The supplementary tables 1 and 3 given in SI. 5 are not sufficient to support the presented data.

Response: Thank you for the constructive suggestion. As suggested, the chemical formula, corresponding elemental analysis with found and calculated mass percentages, molecular weight as well as the yields for CC3 cage, reduced CC3 cage, C-Cage, A-Cage, C-Cage-SS, C-Cage-VB, A-Cage-Imidaz has been provided in the section of 2. Synthetic procedures in supplementary information. One example of CC3 cage is presented here (1st paragraph, page 4 in supplementary information):

“..., then vacuum dried at 80 °C for 24 h. The CC3 cage was obtained as white powder (650 mg, yield: 75%).

¹H-NMR (400 MHz, CDCl₃/CD₃OD): δ 8.17 (s, CH=N, 12H), 7.94 (s, ArH, 12H), 3.40 (m, CH on cyclohexane, 12H), 1.83-1.48 (m, CH₂ on cyclohexane, 48H) ppm; ¹³C-NMR (400 MHz, CDCl₃/CD₃OD): δ 158.88, 135.40, 129.05, 73.52, 31.86, 23.29 ppm; ESI-MS (m/z): [M+H]⁺ calcd. for C₇₂H₈₄N₁₂, 1117.5480; found, 1117.6887; elemental analysis (calcd., found for C₇₂H₈₄N₁₂): C (77.38, 77.30) H (7.58, 7.62), N (15.04, 15.00).”

As for hypercrosslinked polymers C-Cage⁺<PoPIL⁻, A-Cage<PoPIL⁺, [Au<C-Cage⁺]⁻<PoPIL⁻ and [Au<C-Cage⁺&Fe⁺]⁻<PoPIL⁻, it is difficult to define their accurate chemical formulas owing to their radical polymerization essence that normally cause a random and uncontrollable formation of the product. Moreover, the hypercrosslinked polymers are insoluble in organic solvent, their molecular weights are also difficult to determine (a single piece is in fact a single molecule). Normally, FT-IR and elemental analysis as well as ¹³C solid state NMR are employed to confirm the chemical structure and composition for hypercrosslinked polymers. According to the review's comments, to make our results clear and reproducible, the yields and

elemental analysis results for these materials were added in the supplementary information (2. Synthetic procedure and Supplementary Table 2). The Supplementary Table 1, 2 and 4 have also been updated as follows.

Supplementary Table 1. Elemental analysis of the cage molecules and supramolecular assemblies

Entry	C (wt%)		H (wt%)		N (wt%)	
	calcd.	found	calcd.	found	calcd.	found
CC3	77.38	77.30	7.58	7.62	15.04	15.00
RCC3	75.74	75.70	9.53	9.55	14.72	14.66
C-Cage	54.76	54.66	7.66	7.58	10.64	10.73
CC3-OH	73.19	73.12	7.17	7.14	14.23	14.17
A-Cage	66.85	66.79	8.10	8.11	12.99	13.07
C-Cage-SS-12	60.19	60.08	6.13	6.05	5.01	5.17
C-Cage-SS-9	59.46	59.40	6.34	6.25	5.78	5.80
C-Cage-SS-6	58.45	58.39	6.62	6.50	6.82	6.93
C-Cage-VB	74.05	73.91	7.04	7.15	5.76	5.57
A-Cage-imidaz	69.09	69.09	7.89	7.94	19.34	19.42

Supplementary Table 2. Elemental analysis of the hypercrosslinked polymer composites

Entry	C (wt%)	H (wt%)	N (wt%)
C-Cage ⁺ cPoPIL ⁻ -12 (9:1) ^[a]	89.11	7.70	0.61
C-Cage ⁺ cPoPIL ⁻ -12 (3:1) ^[a]	82.53	7.42	1.53
C-Cage ⁺ cPoPIL ⁻ -12 (1:1) ^[a]	75.92	7.04	2.12
C-Cage ⁺ cPoPIL ⁻ -9 (1:1) ^[a]	76.10	7.12	2.74
C-Cage ⁺ cPoPIL ⁻ -6 (1:1) ^[a]	75.54	7.23	3.73
C-Cage ⁺ cPoPIL ⁻ -12 (1:1) ^[b]	83.01	7.29	2.53
A-Cage ⁻ cPoPIL ⁺ -4 (1:1) ^[c]	80.83	7.61	9.33
[Au cC-Cage ⁺] cPoPIL ⁻ ^[d]	75.53	6.78	2.37
[Au cC-Cage ⁺ & Fer ⁺] cPoPIL ⁻ ^[d]	75.57	6.83	2.19

[a] SS as counteranion, the molar ratio of SS to C-Cage varies from 12:1 to 9:1 and 6:1, mass ratio of DVB to C-Cage-SS varies from 9:1 to 3:1 and 1:1.

[b] VB as counteranion, the molar ratio of VB to C-Cage is 12:1, mass ratio of DVB to C-Cage-VB is 1:1.

[c] Imidaz as counteranion, the molar ratio of imidaz to A-Cage is 4:1, mass ratio of DVB to A-Cage-imidaz is 1:1.

[d] SS as counteranion, the molar ratio of SS to C-Cage is 12:1, mass ratio of DVB to C-Cage-SS is 1:1.

Supplementary Table 4. Metal content determined by ICP-OES measurement

Entry	Metal	Content
Au⊂C-Cage ⁺	Au	2.42 wt%
[Au⊂C-Cage ⁺]⊂PoPIL ⁻	Au	0.78 wt%
Au⊂PoPIL ⁻	Au	0.47 wt%
	Au	0.74 wt%
[Au⊂C-Cage ⁺ &Fe ⁺]⊂PoPIL ⁻	Fe	0.44 wt%
Aqueous solution after catalytic reaction ([Au⊂C-Cage ⁺]⊂PoPIL ⁻ catalyst was removed by centrifuge)	Au	< 5 ppm

5. Actually, description of the syntheses should be more detailed in the supporting information section. For instance, in paragraph SI. 2.5, the A-cage formation results from reduction of the CC3-OH cage in the presence of NaBH₄. The procedure consists of solvent removal (methanol, DCM and water) under vacuum. Finally, the residue is washed with water. Is it enough to remove the sodium ions and the boron-containing species? What is the purity of this compound? This point should be cleared and the elemental analysis and yield should be given.

Response: We thank the reviewer for the thoughtful suggestion and comment. We have checked all the synthetic methodologies and made some changes in the description to make the whole synthesis procedures clearer for the readers. Moreover, the elemental analysis and yield have been added in the supplementary information for all the materials. Please see the details in the revised supplementary information.

As for the reduction procedure of CC3 and CC3-OH cage, to remove the by-product (Na⁺ and BO₂⁻) generated from NaBH₄, we used a large amount of water to wash the product (normally 100 mL × 3 times for CC3 reduction and 20 mL × 3 times for CC3-OH reduction is enough). Since the ions are easy to be dissolved in water and BO₂⁻ ion can be readily hydrolyzed to generate OH⁻ base in water solution, so we monitor the pH value of the product suspension to confirm whether the ions were removed completely. When the suspension became neutral, the ions were expected to be totally removed. However, if the pH value of the product suspension is higher than 7, more washing procedure is needed until it becomes neutral. The purity of the compound was confirmed by the ¹H-NMR. The following sentences have been revised in the supplementary information to make the reduction procedure clearer and easier to reproduce.

For CC3 cage reduction (2.2 Synthesis of reduced amine CC3 (RCC3) cage, page 4 in supplementary information):

“The residual was washed with a large amount of water for 3 times (100 mL × 3) to completely remove the decomposed product of NaBH₄. If the pH value of the product suspension is higher than 7, the water-washing procedure is continued until it becomes neutral. The resulting sample was then vacuum dried at 80 °C for 24 h to afford RCC3 as white powder (485 mg, yield: 95%).”

For CC3-OH cage reduction (2.5 Synthesis of reduced CC3-OH anionic cage (A-Cage), page 5 in supplementary information):

“The residual was washed with a large amount of water for 3 times (20 mL × 3) to completely remove the decomposed product of NaBH₄. If the pH value of the product suspension is higher than 7, the water-washing procedure is continued until it becomes neutral. The resulting sample was then vacuum dried at 80 °C for 24 h. A-Cage was obtained as pale yellow powder (50 mg, yield: 91%).”

Reviewer #3 (Remarks to the Author):

I am sorry to be very ambivalent on this manuscript. On one hand, it presents beautiful science. On the other hand, the paper is not easily readable, since it is filled with many methodological details and it presents a discussion that is not always appropriate given the data presented. I appreciate the computational component of the manuscript but this is not the key one.

Response: We thank the reviewer for the thoughtful suggestion and comment. We have thoroughly revised the manuscript and divided the long “Results” section into several themed parts and subtitled them with concise phrase to make the whole manuscript more readable and clearer for the readers.

Reviewer #4 (Remarks to the Author):

In this work, a dual-host supramolecular composite was created through an ion pair-directed supramolecular assembly strategy, which can be used for the immobilization of catalysts (Au cluster or cationic ferrocene). This dual-host molecular hierarchy enables a charge-selective substrate sorting effect to the catalysts, which amplifies their catalytic activity by at least one order of magnitude as compared to catalysts confined by the mono-host. This work is of fundamental importance and novelty. Considering this, it can be accepted after the following major issues are addressed:

1. How can the authors prove that metal clusters are enclosed in cages rather than adsorbed to surfaces? Related discussion and characterization should be provided.

Response: It is a very good point. Generally speaking, characterization of tiny-sized clusters inside a cage is a great challenge. In fact, the encapsulating metal cluster into the same cationic cage (C-Cage) has been well demonstrated by our previous work (*Chem. Sci.* **2019**, *10*, 1450). In that work, we first used common technique such as HAADF-STEM to determinate the size of Au cluster (0.65 ± 0.2 nm) that matches well with the pore size of cage (~ 0.72 nm), indicative of possibility of cage encapsulation. Then, we used the NMR technique to analyze Au@C-Cage⁺ and specifically, the spatial relationship between the cage host and the Au cluster. In comparison to C-Cage, broadening of all peak widths in the ¹H-NMR spectrum of Au@C-Cage⁺ is observed, indicative of a local restricted motion and structural heterogeneity stemming from encapsulation of Au clusters. Similar observations were reported in other metal cluster@cage systems (*J. Am. Chem. Soc.* **2014**, *136*, 1782; *Nat. Catal.* **2018**, *1*, 214), in which the cage shell tightly wrapped around the metal cluster, and experienced restricted mobility and fast spin relaxation. 2D diffusion ordered spectroscopy ¹H-NMR (2D DOSY ¹H-NMR) further showed similar diffusion coefficients for Au@C-Cage⁺ (2.16×10^{-6} cm² s⁻¹) and C-Cage (2.06×10^{-6} cm² s⁻¹), confirming the similar size and shape of native C-Cage and Au@C-Cage⁺. Such observation proves that Au clusters rest inside the cage cavity instead of intercage interactions or aggregation on the cage surface. In addition, the size of Au@C-Cage⁺ determined by cryo-EM data and DLS (consistent with single native C-Cage) excludes the possibility of Au clusters stabilized by multiple cages.

In current work, the Au cluster was encapsulated within the same C-Cage (following our previous procedure) before the supramolecular assembly and co-polymerization process, no aggregation or large particle was observed in final host-in-host materials from the HAADF-STEM measurement, demonstrating that the Au cluster was well confined inside the C-Cage. This result also indicates an ultra-stability of the encapsulated Au cluster.

In the revised manuscript, we have added additional description to make the claim more clarified.

“(The encapsulating of Au cluster into C-Cage has been well demonstrated in our previous work.³⁹)” (2nd paragraph, page 7 in the main text)

2. How many metal clusters are present in the catalyst? How can the authors ensure consistency in the number of active centers in catalytic reactions?

Response: We thank the reviewer for the constructive question. The mass percentages of Au in Au@C-Cage⁺ and [Au@C-Cage⁺]⁻PoPIL⁻ were confirmed by ICP-OES analysis (as shown in Supplementary Table 4). First, the Au clusters in [Au@C-Cage⁺]⁻PoPIL⁻ display a similar average size (0.67 ± 0.17 nm, Fig. 2f, g) to those in Au@C-Cage⁺ (0.65 ± 0.2 nm, data from our previous

work, *Chem. Sci.* **2019**, *10*, 1450), indicative of the strong confinement of cage to maintain the same sized Au cluster after self-assembly and polymerization. Then, the mass percentage of Au in different catalysts was measured by ICP (Supplementary Table 4). Accordingly, the amount of catalyst was used by considering the following equation with aim to ensure the consistency in absolute amount of Au in the catalytic reactions:

$$m_{cat} = \frac{m_{Au}}{c_{Au}}$$

where m_{cat} represents the mass of catalyst, m_{Au} is the mass of Au used in the catalytic reaction and c_{Au} indicates the Au mass percentage in the catalysts that can be obtained from the ICP results.

The similar procedure was used to fix the Fe based catalyst.

For clarity, we have also supplied the mass of catalysts used in catalytic experiment in supplementary information, and the following sentence has been added in the revised manuscript: In the manuscript (1st paragraph, page 8 in the main text):

“..., and the exact metal contents were fixed at the same amount for different catalysts according to the ICP results (Supplementary Table 4).”

In the 3.1.1 Catalytic performance comparison (page 9 in the supplementary information):

“In comparison, the nonconfined Au@C-Cage⁺ catalyst (0.645 mg, calculated according to the ICP results, in Supplementary Table 4) with the same amount of Au as [Au@C-Cage⁺]₂POPIL⁻ catalyst was utilized.”

In the 3.2.3 Activity evaluation of the enzymatic-like TMB oxidation cascade reaction (page 11 in the supplementary information):

“In comparison, a mixture of free Au@C-Cage⁺ (0.612 mg) and Fe³⁺ (50 mM, 3.14 μL, calculated according to the ICP results, in Supplementary Table 4) catalysts with the same Au and Fe content as [Au@C-Cage⁺& Fe³⁺]₂POPIL⁻ catalyst were also evaluated.”

In the 3.2.3 K_m calculation using the Michaelis-Menten equation (page 11 in the supplementary information):

“..., a mixture of free Au@C-Cage⁺ (0.612 mg) and Fe³⁺ (50 mM, 3.14 μL) as well as confined [Au@C-Cage⁺& Fe³⁺]₂POPIL⁻ (2 mg) were used as catalysts,...”

In the 3.2.3 Substrate channelling investigation (page 12 in the supplementary information):

“In comparison, a mixture of free Au@C-Cage⁺ (0.612 mg) and Fe³⁺ (50 mM, 3.14 μL) catalysts with the same Au and Fe content as [Au@C-Cage⁺& Fe³⁺]₂POPIL⁻ catalyst were also evaluated.”

In the 3.2.4 Activity evaluation of the enzymatic-like OPD oxidation cascade reaction (page 12 in the supplementary information):

“In comparison, a mixture of free Au@C-Cage⁺ (0.612 mg) and Fe³⁺ (50 mM, 3.14 μL) catalysts with the same Au and Fe content as [Au@C-Cage⁺& Fe³⁺]₂POPIL⁻ catalyst were also evaluated.”

3. How about the stability of the catalyst? The TEM, ICP, and BET characterizations of the catalysts after cyclic reaction should be provided.

Response: Thanks for your comments. To reveal the stability of the catalyst, N₂ sorption analysis,

TEM measurement and ICP-OES analysis were conducted to characterize the porous structure and size of Au cluster in $[\text{Au@C-Cage}^+]\text{@PoPIL}^-$ catalyst after 5 cycles' catalytic reaction. As shown in the Supplementary Fig. 44, Au clusters did not show any aggregation after 5 cycles' reaction, with an average size of 0.69 ± 0.15 nm, which is comparably the same to the as-prepared one (0.67 ± 0.17 nm). ICP results revealed that negligible metal leaching happened during the reaction (Supplementary Table 4). Moreover, the N_2 gas sorption analysis shows a surface area of around $418 \text{ m}^2\text{g}^{-1}$ (by BET equation), which is close to the as-prepared one ($410 \text{ m}^2\text{g}^{-1}$). In summary, the above-mentioned results provide strong evidences to confirm the high stability of the catalyst. Moreover, these related results have been added into the supplementary information, and the following sentence has been revised in the manuscript for clarity.

“Moreover, the current $[\text{Au@C-Cage}^+]\text{@PoPIL}^-$ catalyst is robust that can be recycled for at least five times without obvious activity loss (Supplementary Fig. 42), and without metal leaching as verified by the ICP test (Supplementary Table 4). The surface area calculated from the BET equation, and the HAADF-STEM measurements indicate the maintained porosity (Supplementary Fig. 43) and size of Au cluster (Supplementary Fig. 44), respectively, after catalytic cycles.” (1st paragraph, page 9)

Supplementary Figure 43. Determination of the specific surface area (according to the BET method) of $[\text{Au@C-Cage}^+]\text{@PoPIL}^-$ after 5 cycles' catalytic reaction.

Supplementary Figure 44. (a) HAADF-STEM image of the Au clusters uniformly dispersed in the

recycled [Au@C-Cage⁺]₂PoPIL⁻ catalyst, scale bar: 5 nm, (b) statistic size distribution histogram of Au clusters (data calculated from 203 counts).

4. Is the reaction efficiency of this tandem catalyst comparable to that of a natural enzyme cascade reaction (such as glucose-HRP system)?

Response: We thank the reviewer for the thoughtful question. We have evaluated the catalytic performance of cascade TMB oxidation experiments by the current artificial catalytic system and natural enzyme (GOx-HRP). Specifically, the confined [Au@C-Cage⁺&Fer⁺]₂PoPIL⁻ catalyst (1.5 μg) was added into a 2 mL solution of glucose (0.05 mM-2.5 mM) and TMB (0.4 mM) mixture. Then, the reaction was initiated by inletting oxygen. The absorbance of the reaction solution at λ_{max}=652 nm was afterwards detected by the UV-Vis spectroscopy with the interval of 5 mins and lasted for 30 mins to calculate the initial reaction velocity. In comparison, a mixture of free GOx (3.33 nM, final concentration in 2 mL solution) and HRP (11.4 nM, final concentration in 2 mL solution) enzymes was also evaluated (Note: the enzymes of GOx (100 U/mg, 150 kDa, obtained from Shanghai Yuanye company) and HRP (> 160 U/mg, 44 kDa, obtained from Aladdin company) were used as received). Then the plot of initial reaction velocity (v) against glucose concentration can be obtained for both catalytic systems (Figure R1a) and fitted by the nonlinear regression of the Michaelis-Menten equation (Figure R1b). The kinetic data including V_{max} (the maximal reaction velocity), K_m (Michaelis constant, representing the substrate binding affinity), k_{cat} (turnover number) and k_{cat}/K_m (specificity constant, representing the enzyme relative rate) for different catalytic systems were obtained according to following equations and shown in Table R1.

As shown in Table R1, [Au@C-Cage⁺&Fer⁺]₂PoPIL⁻ catalyst reveals a comparable activity (76.8 min⁻¹ mM⁻¹) to that of free-GOx-HRP enzyme (70.0 min⁻¹ mM⁻¹). This can be attributed to stronger binding affinity (smaller K_m) to the glucose substrate owing to its nanoconfinement effect, which also demonstrates the unique advantage of the hierarchical “host-in-host” catalyst.

Equations:

$$v = V_{max} \times \frac{C_{glucose}}{K_m + C_{glucose}}$$

$$k_{cat} = \frac{V_{max}}{C_{cat}}$$

where V_{max} is the maximum reaction velocity, C_{glucose} is the concentration of glucose, K_m is the Michaelis constant, k_{cat} is the turnover number and C_{cat} is the concentration of catalyst.

Figure R1. (a) Comparison of initial reaction velocity and (b) linear fitting of double reciprocal

plots for confined $[\text{Au@C-Cage}^+ \& \text{Fer}^+] \subset \text{PoPIL}^-$ catalyst and a mixture of free GOx-HRP enzyme against different glucose concentration.

Table R1. Kinetic data for different tandem catalytic systems

	V_{\max} ($\mu\text{M min}^{-1}$)	K_m (mM)	k_{cat} (min^{-1})	k_{cat}/K_m ($\text{min}^{-1} \text{mM}^{-1}$)
$[\text{Au@C-Cage}^+ \& \text{Fer}^+] \subset \text{PoPIL}^-$	1.29	0.60	46.1	76.8
Free GOx-HRP enzyme	0.74	3.17	222.2	70.0

5. Please check Figure 5b and Supplementary Figure 55.

Response: We thank the reviewer for the useful suggestion. The figures have been updated in the revised manuscript and supplementary information for clarity.

Fig. 5 (b) UV-Vis spectra for TMB oxidation cascade reaction by confined $[\text{Au@C-Cage}^+ \& \text{Fer}^+] \subset \text{PoPIL}^-$ catalyst and a mixture of free Au@C-Cage^+ and Fer^+ catalysts in the solution.

Supplementary Figure 61. (a) UV-Vis spectra of the OPD oxidation cascade reaction by confined $[\text{Au@C-Cage}^+ \& \text{Fer}^+] \subset \text{PoPIL}^-$ catalyst and a mixture of free Au@C-Cage^+ and Fer^+ catalysts in the

solution.

6. In Figure 5e, the reduced catalytic activity may be attributed to the toxicity of the catalytic central site rather than the consumption of intermediates of H_2O_2 . More detailed experimental evidence needs to be supplemented.

Response: We thank the reviewer for the thoughtful suggestion and comment.

In most conditions, the poisoning of catalyst actually may cause a large efficiency decrease. However, in our work, we stated that the reduced catalytic activity should be attributed to the consumption of the intermediate of H_2O_2 , which further contributed to the oxidation of TMB substrate. To verify our statement, the following experiments were designed and conducted (Note: all the experiments were conducted at 298 K).

Firstly, we have confirmed that in the catalyst-free condition, the thiourea can react with H_2O_2 through the simple redox mechanism (Scheme R1, *ACS Chem. Biol.* **2017**, *12*, 1737). Specifically, H_2O_2 (100 mM, 10 μL , 5eq) was directly added into the thiourea solution (0.1 mM, 2 mL, 1eq). The reaction process was evaluated by monitoring the concentration of thiourea using the UV-Vis spectroscopy. As shown in the Figure R2a, the absorption at 236 nm that was assigned to thiourea (according to *ACS Chem. Biol.* **2017**, *12*, 1737) gradually decrease in the presence of H_2O_2 . Meanwhile, a new absorption at 265 nm appeared accordingly that can be attributed to the generation of thiourea dioxide product (*Carbohydr. Polym.* **2007**, *67*, 66), indicating that the reaction occurs between thiourea and H_2O_2 . In comparison, no absorption changes without H_2O_2 (Figure R2b).

Scheme R1. Reaction equation of thiourea oxidation by H_2O_2

Figure R2. UV-Vis spectra of thiourea (a) with and (b) without the addition of H_2O_2 oxidant. The concentrations of thiourea and H_2O_2 are 0.1 mM and 0.5 mM, respectively. The letter “A” in Figure R2a represents the absorption of thiourea substrate, and the letter “B” represents the absorption of thiourea dioxide product.

Secondly, we demonstrated that the catalytic activity of confined $[\text{Au}@\text{C-Cage}^+\&\text{Fer}^+]\text{C-PoPIL}^-$ catalyst still retained at a high efficiency after the adsorption of thiourea. The adsorption experiment was conducted by immersing $[\text{Au}@\text{C-Cage}^+\&\text{Fer}^+]\text{C-PoPIL}^-$ catalyst (2mg) into thiourea solution (0.5 mM, 2 mL) for 15 min, which is the same condition as the thiourea

inhibitor challenging the substrate channelling experiments presented in the manuscript. After the complete diffusion of thiourea, the $[\text{Au@C-Cage}^+\&\text{Fe}^+]\text{@PoPIL}^-$ was isolated from the solution by centrifuge. The adsorption amount was estimated by measuring the concentration change of thiourea solution from the UV-Vis spectroscopy. As shown in Figure R3a, only a tiny amount of thiourea (approximately 3%) was trapped by $[\text{Au@C-Cage}^+\&\text{Fe}^+]\text{@PoPIL}^-$. This indicated that thiourea may go through the porous catalyst matrix by substrate diffusion rather than largely adsorbed into the network. Then, the $[\text{Au@C-Cage}^+\&\text{Fe}^+]\text{@PoPIL}^-$ catalyst with a tiny amount of thiourea was directly applied to the TMB oxidation cascade reaction. UV-Vis spectroscopy was applied to evaluate the catalytic activity. As shown in Figure R3b, c, the $[\text{Au@C-Cage}^+\&\text{Fe}^+]\text{@PoPIL}^-$ catalyst accommodating a tiny amount of thiourea within the pore shows comparable activity ($0.4436 \mu\text{M min}^{-1}$) to that of pristine $[\text{Au@C-Cage}^+\&\text{Fe}^+]\text{@PoPIL}^-$ catalyst ($0.4449 \mu\text{M min}^{-1}$, Supplementary Fig. 60a), with negligible activity loss of 0.3 %. However, in the main text, we use the thiourea inhibitor to challenge the channelling experiments, *i.e.* 2mg of $[\text{Au@C-Cage}^+\&\text{Fe}^+]\text{@PoPIL}^-$ catalyst was added into thiourea solution (0.5 mM, 2 mL) and reacted for 15 min, the rate constant is $0.3974 \mu\text{M min}^{-1}$, (Supplementary Fig. 60b), dropped by 10.7 % as compared to that of pristine catalyst without thiourea (Figure R3d). These experiments indicating that the thiourea mainly contacts with H_2O_2 intermediate through the substrate diffusion effect across the pore rather than adsorbed on the surface of metal sites.

In summary, we can conclude that the reduced catalytic activity of $[\text{Au@C-Cage}^+\&\text{Fe}^+]\text{@PoPIL}^-$ catalyst after the addition of thiourea competing reagent is due to the consumption of intermediates of H_2O_2 instead of the toxicity of the catalytic sites.

Figure R3. (a) UV-Vis spectra of thiourea solution before and after adsorption by $[\text{Au@C-Cage}^+\&\text{Fe}^+]\text{@PoPIL}^-$ catalyst, both of the solutions were diluted for 5 times before measurement, (b) UV-Vis spectra and (c) linear fitting of reaction rate constant for the TMB oxidation cascade

reaction by $[\text{Au}@\text{C-Cage}^+ \& \text{Fe}^+]@\text{PoPIL}^-$ catalyst after adsorption of a tiny amount of thiourea, (d) comparison of the reaction rate constant decrease in thiourea inhibitor challenging the substrate channelling experiments conducted in the main text (I) and the current designed experiments (II).

7. *Fig. 2f is recommended to be updated because of the low magnification.*

Response: We thank the reviewer for the useful suggestion. The TEM image of the Au clusters with a higher magnification has been updated in the Fig.2f in the revised manuscript.

REVIEWER COMMENTS

Reviewer #2 (Remarks to the Author):

The manuscript submitted by Yuan et al. has been positively evaluated by four reviewers who requested many constructive major revisions. Considering that the authors have fulfilled all the requested points, remarks, and suggestions and even provided missing experimental data raised by the reviewers, the resulting revised manuscript of Yuan et al. reaches now the high-level standard required by Nature Comm. The Editor can trust within the real added-value of this manuscript arising from the virtuous circle that includes the four reviewers, the authors and the Editorial board. Congrats to the team!

Reviewer #4 (Remarks to the Author):

The authors have revised the manuscript according to reviewers' comments and it is recommended to be accepted at the current stage.

Point-by-point response to reviewers' comments

Reviewer #2 (Remarks to the Author):

The manuscript submitted by Yuan et al. has been positively evaluated by four reviewers who requested many constructive major revisions. Considering that the authors have fulfilled all the requested points, remarks, and suggestions and even provided missing experimental data raised by the reviewers, the resulting revised manuscript of Yuan et al. reaches now the high-level standard required by Nature Comm. The Editor can trust within the real added-value of this manuscript arising from the virtuous circle that includes the four reviewers, the authors and the Editorial board. Congrats to the team!

Response: We are grateful for your constructive comments and suggestions to improve the quality of our work.

Reviewer #4 (Remarks to the Author):

The authors have revised the manuscript according to reviewers' comments and it is recommended to be accepted at the current stage.

Response: We thank the reviewer for the reviewing of our manuscript again.